# The Modified Broström Procedure with Suture-Tape Augmentation for Chronic Lateral Ankle Instability

**DOI:** 10.3390/jcm14051683

**Published:** 2025-03-02

**Authors:** Byung-Ki Cho, Sung-Hoo Kim

**Affiliations:** 1Department of Orthopaedic Surgery, Chungbuk National University Hospital, Cheongju 28644, Republic of Korea; hoo414414@hanmail.net; 2Department of Orthopaedic Surgery, College of Medicine, Chungbuk National University, Cheongju 28644, Republic of Korea

**Keywords:** ankle, chronic instability, modified Broström procedure, suture-tape augmentation

## Abstract

**Background/Objectives**: As a representative anatomic ankle ligament repair technique, the Broström procedure continues to be modified to reach better clinical outcomes, superior mechanical stability, early rehabilitation, and minimal risk of recurrent instability. This study aimed to evaluate the intermediate-term clinical outcomes after the modified Broström procedure (MBP) with suture-tape augmentation for chronic lateral ankle instability. **Methods**: Ninety-four patients with chronic lateral ankle instability were followed for ≥3 years after MBP augmented with suture tape. The patient-reported clinical outcomes were evaluated with the Foot and Ankle Outcome Score (FAOS) and the Foot and Ankle Ability Measure (FAAM). The changes in mechanical ankle stability were evaluated with physical examination and periodic stress radiography. The changes in static and dynamic postural control ability were assessed with the single-leg stance test and Biodex posturography. **Results**: FAOS and FAAM scores significantly improved from preoperative means of 52.6 and 54.2 points to 91.8 and 90.5 points at final follow-up, respectively (*p* < 0.001). Talar tilt angle and anterior talar translation significantly improved from preoperative means of 15.4° and 14.3 mm to 2.7° and 4.5 mm at final follow-up, respectively (*p* < 0.001). Two patients (2.1%) complained of a recurrence of mechanical and functional instability. One patient (1.1%) showed non-specific inflammation related to a suture tape. Balance retention time significantly improved from a preoperative mean of 3.7 to 6.4 s at final follow-up (*p* < 0.001), with a non-significant side-to-side difference. The overall stability index significantly improved from a preoperative mean of 3.7 to 1.9 at final follow-up (*p* < 0.001), with a significant side-to-side difference. **Conclusions**: The MBP augmented with suture tape appears to be an effective surgical technique for chronic lateral ankle instability. Through anatomic repair of attenuated ankle ligaments and suture-tape augmentation, this modified procedure can provide reliable stability and minimal risk of recurrent instability. In addition, static and dynamic postural control ability may be improved through continuous proprioceptive-oriented rehabilitation following surgery.

## 1. Introduction

The original Broström procedure for lateral ankle instability was a direct anatomical repair that involved suturing the torn or attenuated ligament ends back together. The Broström procedure has since evolved with a variety of modifications. One of the representative modified Broström repair techniques was the Broström–Gould procedure, which included augmentation with the inferior extensor retinaculum over the repaired ligaments. Another technique was the Broström–Karlsson procedure, which made shortening of the ligaments themselves through reattachment using transosseous tunnels [1]. As a mainstay in operative treatment for patients with chronic lateral ankle instability, the modified Broström procedure (MBP) has long demonstrated satisfactory clinical and functional outcomes through anatomical ligament repair. Although the MBP yields a restoration of ankle stability by anatomical repair of the affected ligament tissues, patients with questionable remnant tissue quality for repair are frequently encountered. With controversy regarding the relatively poor prognostic factors for the conventional MBP [1,2,3], this surgical technique is consistently being modified to achieve superior clinical outcomes, reliable stability, less complication, early rehabilitation, and rapid return to activities. As one of the recently emerging techniques, suture-tape augmentation, in addition to the conventional Broström repair, has been developed in order to provide increased mechanical stability and protection to the repaired lateral ligaments of the ankle joint [2,4,5,6]. Various biomechanical analyses have demonstrated that MBP augmented with suture tape is superior to the MBP alone and equivalent to the intact native lateral ligaments of the ankle [7,8,9].

With the increasing practical use of suture-tape augmentation, its clinical benefits and potential complications have been actively discussed by a large number of surgeons. To date, many case series reported this technique showed satisfactory clinical outcomes and fast return to sports activity at short-to-intermediate-term follow-up [4,5,10,11,12]. Recent literature supports the use of suture-tape augmentation in various populations, including patients who are relatively contraindicated for conventional MBP [2,6,13,14,15,16]. However, there is ongoing debate on usefulness in CAI patients with poor prognostic factors of the anatomical ligament repair (MBP) [1,2,3]. In addition, there is still a wide variability between the published studies in terms of study design, patient selection, methodology, surgical indication, and techniques.

We hypothesized that the MBP augmented with suture tape would provide advanced mechanical stability as well as the advantages of anatomical ligament repair. In addition, this combined procedure may be an effective surgical option for chronic ankle instability with poor prognostic factors related to the conventional MBP. Given this background, we investigated the intermediate-term clinical outcomes after the MBP with suture-tape augmentation for chronic lateral ankle instability.

## 2. Materials and Methods

Study subjects

Between October 2017 and November 2021, a total of 119 patients (122 ankles) consecutively underwent the MBP with suture-tape augmentation for chronic lateral ankle instability. All patients underwent at least 3 months of failed rehabilitation before surgical treatment. The MBP augmented with suture tape was limitedly indicated for the selected patients as follows: 21 patients with generalized ligamentous laxity (Beighton score ≥ 5 points) [17], 11 patients with failed previous ligament repair surgery, 27 patients with poor quality of the remnant lateral ligamentous tissue as shown by preoperative magnetic resonance imaging (MRI), 39 patients with insufficient repairable ligamentous tissue at intraoperative inspection, 12 patients with large subfibular ossicle (diameter ≥ 5 mm), and 9 patients with hyper-obesity (body mass index ≥ 30 kg/m^2^). Of these patients, 98 patients (98 ankles) eligible for inclusion criteria were enrolled in this study. Finally, 94 patients (94 ankles) with a follow-up longer than 3 years after operation were analyzed (Figure 1). The inclusion criteria for this study were as follows: (1) unilateral ankle instability, (2) no advanced osteochondral lesion needing microfracture or autologous osteochondral transplantation, and (3) no previous ankle ligament surgery history. With consideration for heterogeneity in details of the surgical technique (range of skin incision, number and ingredient of remaining suture anchors, insertion point of suture anchors, use of the calcaneofibular ligament or inferior extensor retinaculum, etc.) and remnant ligamentous tissue condition (presence of iatrogenic scar tissue adhesion and remained suture materials), we determined a history of previous ankle ligament surgery as one of the exclusion criteria. In this study, 3 patients with bilateral ankle instability, 6 patients with concomitant microfracture or osteochondral transplantation, and 11 patients with previous ankle ligament surgery were excluded. In addition, one patient was unwilling to participate and was excluded from this study.

The mean age of the patients was 29.6 years (range, 19 to 45 years), and the mean duration of symptoms was 40.2 months (range, 8 to 92 months). This study included 51 male and 43 female patients. For this study, we prospectively collected patient data and retrospectively analyzed it. All evaluation factors were determined during the study design phase, and patients with eligibility for inclusion criteria were enrolled consecutively. The overall research protocol was approved by the ethics committee of the Institutional Review Board in September 2017 (prior to the enrollment of the first participant who underwent an operation). Informed consent regarding the promised visit frequency and examination content after surgery was given to all participants.

2Operative procedure and rehabilitation protocol

Following the concomitant simple arthroscopic procedures, the modified Broström–Gould procedure was performed in the conventional manner. To reattach the remnant of anterior talofibular (ATF) and calcaneofibular (CF) ligaments, two metallic suture anchors were inserted at the anteroinferior border of the lateral malleolus (Figure 2). Subsequently, a 3.5 mm SwiveLock^®^ biocomposite suture anchor (Arthrex, Naples, FL, USA) with 2 mm width FiberWire^®^ suture tape (Arthrex, Naples, FL, USA) was inserted into the lateral aspect of two metallic anchors. With the ankle joint in neutral flexion and slight eversion, the capsule and lateral ligaments were securely fixed to the fibula (Figure 3). Maintaining the ankle in a neutral position to prevent overtightening of suture tape, the repaired ATFL was augmented with suture tape using a 4.75 mm SwiveLock^®^ anchor fixation into the talar neck (Figure 4). Finally, the inferior extensor retinaculum was advanced and imbricated over the repaired lateral ligaments and suture tape. In summary, the surgical technique used in this study was an anatomical repair of both ATFL and CFL and additional suture-tape augmentation for the repaired ATFL.

A short-leg splint and partial weight-bearing ambulation with crutches were maintained for 3 weeks after surgery. Thereafter, range-of-motion (ROM) exercises and tolerable weight-bearing ambulation with an elastic ankle bandage were permitted. From 4 weeks postoperatively, full weight-bearing gait, peroneal muscle strengthening exercise, and proprioception training were encouraged. A return to sports activities was permitted after at least 8 weeks.

3Evaluation of patient-reported clinical outcomes

Clinical outcomes were periodically assessed with the Foot and Ankle Outcome Score (FAOS) [18] and Foot and Ankle Ability Measure (FAAM) [19], which had been validated to the outcome measures for ankle ligament injury. The FAOS comprises 42 questions and 5 subscales evaluating pain, other symptoms, activities of daily living, sports activities, and quality of life. The FAAM comprises 2 subscales assessing activities of daily living (21 questions) and sports activities (8 questions).

4Evaluation of mechanical ankle stability

Mechanical ankle stability was periodically assessed with physical examination (by one senior surgeon) and stress radiographs. The talar tilt angle and the degree of anterior talar translation were measured on varus and anterior drawer ankle stress radiographs using the Telos device (Telos GmbH, Marburg, Germany) with a consistent load (150 N). All radiological measurements were independently performed by three researchers on a digital PACS imaging system.

5Evaluation of static postural control ability

Static postural control ability was periodically measured using the modified Romberg test (single-leg stance test with eyes closed) [20]. All subjects were instructed to stand on one leg (without shoes) and remain as still as possible with arms outspread and eyes closed. The same researcher assessed a balance retention time before subjects placed their non-stance foot on the floor with a feeling unable to maintain this position. All tests were repeated two times on each leg and the measurements were averaged.

6Evaluation of dynamic postural control ability

Dynamic postural control ability was periodically assessed with Biodex posturography (Biodex Medical Systems, Shirley, NY, USA). All subjects were instructed to maintain their one-leg standing posture with eyes open during the automatic change in platform tilting and rotation from level 8 (most stable) to level 1 (most unstable) for 20 s. Under the supervision of the same researcher, this task was repeated two times on each leg and the measurements were averaged. All tests were performed by first examining the unaffected side and then the affected side. As the postural stability parameters, anterior–posterior stability index (APSI), medial–lateral (MLSI) stability index, and overall stability index (OSI) were obtained. The deficit ratio of dynamic postural control ability relative to the unaffected side was analyzed and multiplied by 100 to describe a percentage (%).

7Statistical analysis

The statistical analysis was performed with SPSS 21.0 (SPSS Inc, IBM Company, Chicago, IL, USA), and a *p*-value ≤ 0.05 with a confidence interval of 95% was set as the statistical level of significance. Wilcoxon signed-rank test was used to compare the changes in patient-reported clinical outcomes, mechanical ankle stability, and static and dynamic postural control ability between affected (operated) and unaffected sides.

## 3. Results

Patient-reported clinical outcomes

FAOS significantly improved from a preoperative mean of 52.6 points (range, 41–68 points) to 91.8 points (range, 81–98 points) at final follow-up (*p* < 0.001). Through comparisons in each subscale, there were statistical differences in all five subscales between preoperative and final FAOS (Table 1). FAAM score significantly improved from a preoperative mean of 54.2 points (range, 44–69 points) to 90.5 points (range, 79–99 points) at final follow-up (*p* < 0.001). There were statistical differences in all two subscales between preoperative and final FAAM scores.

2.Postoperative complications

There were two cases (2.1%) of superficial wound infection, two cases (2.1%) of delayed wound healing, one case (1.1%) of damage to the superficial peroneal nerve, one case (1.1%) of skin irritation by suture materials, two cases (2.1%) of recurrent ankle instability, and one case (1.1%) of nonspecific inflammation related to a suture tape (Table 2). All cases with superficial wound infection or delayed wound healing were treated with intravenous antibiotics and continuous dressing care. Tingling and numbness on the foot dorsum, caused by superficial peroneal nerve injury, spontaneously improved 6 months postoperatively. One case of skin irritation by suture materials underwent suture (vicryl) knot removal under local anesthesia at 8 months postoperatively. One case with recurrent ankle instability underwent revision surgery (lateral ligament reconstruction using all-tendon graft) at 23 months postoperatively, and the other case refused a reoperation despite partial limitation in participation in sports activities. One case of nonspecific inflammation around the incision scar was diagnosed as a foreign-body reaction related to the suture tape and successfully treated with a local steroid (triamcinolone) injection.

3.The changes in mechanical ankle stability

On periodic stress radiographic examination, the talar tilt angle significantly improved from a preoperative mean of 15.4° (range, 7–26°) to a mean of 2.7° (range, 1–7°) at final follow-up (*p* < 0.001). Anterior talar translation significantly improved from a preoperative mean of 14.3 mm (range, 6–20 mm) to a mean of 4.5 mm (range, 3–8 mm) at final follow-up (*p* < 0.001) (Table 3). Although preoperative side-to-side comparisons in talar tilt and anterior talar translation demonstrated significant differences, these returned to non-statistical differences at final follow-up (*p* = 0.461, 0.706, respectively).

4.The changes in static postural control ability

On the periodic modified Romberg test, balance retention time (period to maintain single-leg stance with eyes closed) significantly improved from a preoperative mean of 3.7 s (range, 1–8 s) to 6.4 s (range, 3–14 s) at final follow-up (*p* < 0.001) (Table 4). Although preoperative side-to-side comparison in balance retention time demonstrated a significant difference, this returned to a non-significant difference at final follow-up (*p* = 0.105).

5.The changes in dynamic postural control ability

On periodic posturographic examination, the overall stability index significantly improved from a preoperative mean of 3.7 (range, 2.8–5.2) to 1.9 (range, 0.8–3.1) at final follow-up (*p* < 0.001) (Table 5). As compared to the unaffected side, the deficit ratio of the overall stability index was 59.5% preoperatively, with a significant side-to-side difference (*p* < 0.001). Side-to-side comparison in the overall stability index at final follow-up demonstrated a significant difference (*p* = 0.014), and the deficit ratio of the overall stability index was 26.3%.

## 4. Discussion

The most important finding of this study is that MBP with suture-tape augmentation is an effective surgical option for chronic lateral ankle instability with poor prognostic factors related to conventional anatomical ligament repair. Considering an issue on cost-effectiveness of additional new device, suture-tape augmentation following the anatomical ligament repair was limitedly applied to the patients with generalized ligamentous laxity, prior unsuccessful ligament repair surgery, poor quality of remnant ligamentous tissue, insufficient repairable tissue at intraoperative inspection, large subfibular ossicle, and hyper-obesity (BMI ≥ 30 kg/m^2^). On the intermediate-term follow-up of a mean of 5.4 years, the MBP augmented with suture tape demonstrated satisfactory clinical outcomes and minimal risk of recurrent instability. Another interesting finding of this study is the progressive improvements in static and dynamic postural control ability over time. Compared to the early postoperative period, both parameters showed marked improvements at final follow-up. We recommended all patients to have the supervised proprioception training by a physical therapist for at least 2 months after surgery. Although continuous proprioceptive-oriented exercise was encouraged during every follow-up visit, compliance with rehabilitation treatment might be critically varied among the patients. Nevertheless, this finding suggests that postural control ability may be improved through continuous proprioceptive-oriented rehabilitation following surgery.

There are concerns about the strength and durability of traditional lateral ligament repair (MBP) for chronic ankle instability based on biomechanical load-to-failure studies [7,21]. In a biomechanical study by Waldrop et al., the reconstruction surgery of the anterior talofibular ligament (ATFL) was not able to restore the strength of native ATFL even with ligament repair using suture anchor or traditional Broström procedure [7]. Suture-tape augmentation, in addition to the Broström procedure with a suture anchor, demonstrated a 95% higher torque at failure compared to the traditional Broström procedure and 54% higher torque at failure compared to the Broström procedure with a suture anchor. Early range of motion in postoperative rehabilitation is well known to be helpful for biological healing following ligament repair surgery [22]. A more aggressive rehabilitation program for a fast return to daily and sports activities seems to be recommended to ordinary patients as well as athletes. However, there is still a concern regarding the re-sprain injury and recurrence of instability. Kirk et al. reported that unprotected mobilization following the traditional Broström repair caused the lengthening of a mean of 20% in the ATFL [21]. Excessive elongation of the lateral ligaments during rehabilitation, including early mobilization after MBP, is likely to result in joint laxity and subsequent failure of the repair [7]. Therefore, this concern has raised the need for a new device or technical modification that provides higher stability than the traditional MBP [4,10,23,24,25,26].

Suture-tape augmentation, in addition to the conventional Broström repair, has been developed in order to increase the mechanical stability of the repaired lateral ligaments and tolerance for early functional rehabilitation [2,4,6,10]. Suture tape plays a role as a synthetic ligament which is made of long-chain ultrahigh-molecular-weight polyethylene [2]. Based on the biomechanical superiority of the MBP augmented with suture tape, this technique has been an increasingly popular option for chronic ankle instability [2,27]. The suture-tape augmentation technique has been reported to allow early mobilization and encourage regeneration of the repaired ligament by protecting potential elongation during the ligament healing phase as an internal stabilizer [5,10]. Viens et al. reported that MBP with suture-tape augmentation was similar to intact ATF ligament on ultimate load to failure and stiffness [28]. Schuh et al. reported that MBP augmented with suture tape was biomechanically superior to MBP alone [8]. However, it still remains unclear whether the biomechanical superiority of the MBP augmented with suture tape leads to significantly better clinical outcomes.

Due to the inherent connective tissue abnormality, generalized ligamentous laxity has been generally known to be a poor prognostic factor for anatomic ligament repair [29,30,31]. Park et al. have reported that generalized ligamentous laxity is an independent predictor of poor outcomes with high rates of clinical failure [30]. Xu et al. suggested that additional augmentation procedures or ligament reconstruction using tendon grafts should be considered as primary surgery for patients with high Beighton scores (≥7 points) [31]. Based on the hypothesis that synthetic ligaments, such as suture tape, could overcome these biological issues, Cho et al. investigated the clinical outcomes after suture-tape augmentation for chronic ankle instability with generalized ligamentous laxity (joint hypermobility) and reported that suture-tape augmentation, in addition to the MBP, appeared to be an effective operative alternative for patients with inherent connective tissue deficiency [15].

In regard to the clinical results between modified Broström procedures with and without suture-tape augmentation, a recent systematic review reported that MBP augmented with suture tape showed good short-term clinical outcomes with few complications, comparable to the MBP alone [27]. Xu et al. reported that MBP with suture-tape augmentation achieved a better patient-reported clinical outcome (FAAM) than MBP alone [11]. A multicenter, prospective, randomized study by Kulwin et al. reported that MBP augmented with suture tape provided an earlier return to preinjury level of activity (mean 13.3 vs. 17.5 weeks) and fewer complications (1.7 vs. 8.5%) compared to the MBP alone [10]. Coetzee et al. suggested that MBP with suture-tape augmentation safely allowed for an accelerated rehabilitation protocol, including immediate weight-bearing and early range of motion [5]. Kim et al. reported that MBP augmented with suture tape yielded satisfactory clinical and radiological outcomes in CAI patients with poor-quality remnant ligamentous tissue at intermediate-term follow-up [6]. Because the MBP provides restoration of ankle stability by anatomical repair of the attenuated lateral ligaments, the ability to yield consistent mechanical stability regardless of the remnant ligamentous tissue may be one of the strengths of this technique. In this study, the improvements in clinical and radiographic parameters were consistent with previously published studies. Conversely, MBP with suture-tape augmentation failed to result in superior clinical outcomes to MBP without suture-tape augmentation in a few comparative studies [11,32,33]. Despite excellent results in multiple clinical and radiographic assessment measures, another recent systematic review has reported that there is minimal evidence to support better functional outcomes or lower recurrence rates compared to the MBP alone [13]. A definitive conclusion regarding the effects of additional suture-tape augmentation with the MBP on long-term outcomes needs to be studied with further randomized controlled trials.

The overall complication rate following the MBP with suture-tape augmentation has been reported to be relatively low and comparable to conventional MBP [13,27]. A few systematic reviews reported 5.7–7% of the pooled complication rate for patients undergoing suture-tape internal bracing [13,27]. Although this study showed similarly common complications that had been reported in previous literature, a potential complication such as nonspecific inflammation (foreign body reaction) related to a suture tape should be considered as the ongoing debate. In addition, there is no consensus regarding the most effective and safe protocol for rehabilitation following the MBP with suture-tape augmentation. Further randomized studies with a high level of evidence need to be performed to set a standard rehabilitation strategy, including the timing of weight-bearing and the use of a brace or cast.

This study has some limitations. First, this study included no analysis regarding the time to return to a preinjury level of sports activity. Although fast rehabilitation and early return to sports activity have been reported as a considerable strength of the MBP with suture-tape augmentation [5,32,34], the study design of this prospective case series did not include it as one of the evaluation factors. Second, a question remains whether the increased mechanical stability by suture-tape augmentation results in any difference in the biological healing of the repaired ligaments. Due to the absence of postoperative histologic analysis or routine MRI assessment to evaluate ligament regeneration in this study, the effects of suture-tape augmentation on the healing process following ligament repair are still unclear. Third, the details of surgical techniques (i.e., the best entry point for suture tape, intra- or extra-articular placement of suture tape, proper number of suture anchors) for the MBP with suture-tape augmentation have not yet been standardized. The variability among the surgical techniques may lead to heterogeneous results. Although an internal brace for the ATFL needs to be placed at the anatomic insertion sites on the talus and fibula, there is still a paucity of literature on clinical relevance according to the technical divergence. Further biomechanical or cadaveric studies regarding the best entry point for suture tape and anchors need to be performed. Finally, the lack of a formal comparison group is another limitation of this study. A definitive conclusion regarding the usefulness of MBP with suture-tape augmentation in patients with relatively poor prognostic factors should be answered through a randomized controlled study with more high-level evidence.

## 5. Conclusions

The MBP augmented with suture tape appears to be an effective surgical technique for chronic lateral ankle instability. Through anatomic repair of attenuated ankle ligaments and suture-tape augmentation, this modified procedure can provide reliable mechanical stability and minimal risk of recurrent instability. In addition, static and dynamic postural control ability may be improved through continuous proprioceptive-oriented rehabilitation following surgery.

## Figures and Tables

**Figure 1 jcm-14-01683-f001:**
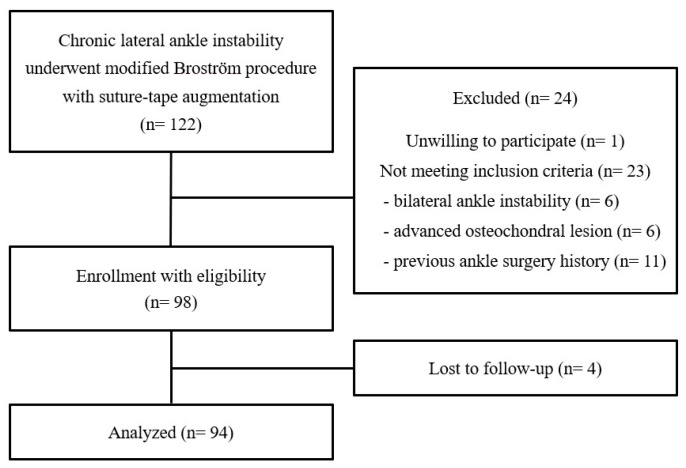
Flowchart diagram of this study.

**Figure 2 jcm-14-01683-f002:**
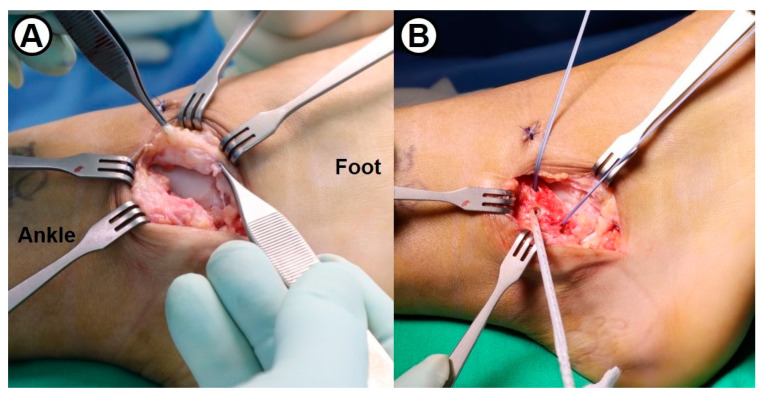
(**A**) Intraoperative photograph showing the attenuated lateral ligamentous tissue in a 34-year-old male patient. (**B**) Following fixation of two 3.5 mm metallic suture anchors into the lateral malleolus, FiberWire^®^ suture tape is secured using a 3.5 mm SwiveLock^®^ biocomposite suture anchor.

**Figure 3 jcm-14-01683-f003:**
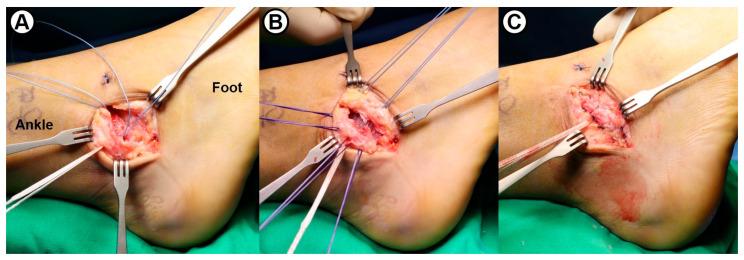
Intraoperative photographs showing (**A**) passage of suture tape outside of the fibular periosteum, (**B**) imbrication with horizontal mattress suture technique, and (**C**) repair of the lateral ligaments and capsule.

**Figure 4 jcm-14-01683-f004:**
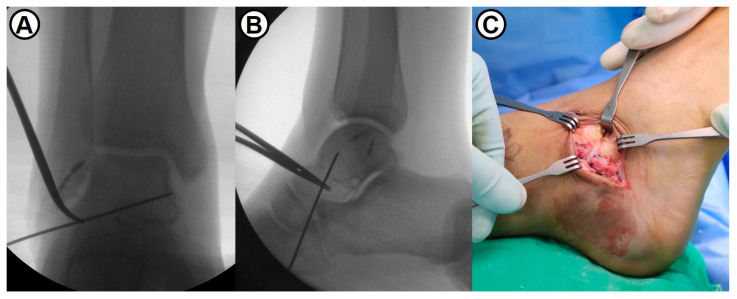
(**A**,**B**) Intraoperative fluoroscopic images show a proper entry-point of suture tape using temporary guidewire fixation. (**C**) The repaired lateral ligaments are augmented with suture tape which is fixed into the talar neck using a 4.75 mm SwiveLock^®^ biocomposite suture anchor.

**Table 1 jcm-14-01683-t001:** Patient-reported clinical outcomes (Wilcoxon signed-rank test).

Subscales	Preoperative	PO 6 Months	PO 1 Year	Final F/U	*p*-Value ^†^
FAOS *					
Pain	67.6 ± 12.8	84.4 ± 10.1	90.2 ± 9.3	93.5 ± 6.4	<0.001
Symptoms	58.2 ± 15.1	85.3 ± 9.8	88.6 ± 9.6	94.4 ± 5.5	<0.001
Activity of daily living	55.8 ± 16.4	84.1 ± 10.4	89.1 ± 10.2	92.8 ± 6.7	<0.001
Sports	31.3 ± 18.6	67.5 ± 16.2	81.3 ± 13.7	85.3 ± 12.1	<0.001
Quality of life	50.1 ± 17.9	84.6 ± 10.5	88.4 ± 10.9	93.1 ± 6.6	<0.001
Total FAOS	52.6 ± 17.2	81.2 ± 13.6	87.5 ± 11.4	91.8 ± 8.1	<0.001
FAAM *					
Daily activity	69.1 ± 15.5	88.5 ± 9.6	91.1 ± 8.8	94.1 ± 5.9	<0.001
Sports activity	39.3 ± 19.4	72.3 ± 15.9	80.5 ± 14.7	86.9 ± 12.6	<0.001
Total FAAM score	54.2 ± 16.8	80.4 ± 13.7	85.8 ± 12.9	90.5 ± 9.4	<0.001

Abbreviations: FAOS, Foot and Ankle Outcome Score; FAAM, Foot and Ankle Ability Measure; PO, postoperative; F/U, follow-up. * Data are represented as scores (mean ± standard deviation) changed on the basis of 100 points. ^†^ Comparison between preoperative and final follow-up.

**Table 2 jcm-14-01683-t002:** Complications after modified Broström procedure with suture-tape augmentation.

	Number (%)
Superficial wound infection	2 (2.1%)
Delayed wound healing	2 (2.1%)
Superficial peroneal nerve injury	1 (1.1%)
Skin irritation by suture materials	1 (1.1%)
Recurrence of ankle instability	2 (2.1%)
Nonspecific inflammation by suture tape	1 (1.1%)

**Table 3 jcm-14-01683-t003:** The changes in mechanical ankle stability (Wilcoxon signed-rank test).

Stress Radiographs	Preoperative	PO 6 Months	PO 1 Year	Final F/U	*p*-Value ^†^
Talar tilt angle (°) *					
Affected (operated) side	15.4 ± 6.8	2.2 ± 1.3	3.1 ± 1.6	2.7 ± 1.4	<0.001
Unaffected side	2.7 ± 1.5	2.9 ± 1.6	2.5 ± 1.3	3.1 ± 1.7	0.509
*p*-value ^‡^	<0.001	0.128	0.354	0.461	
Anterior talar translation (mm) *					
Affected (operated) side	14.3 ± 5.9	3.8 ± 2.1	4.3 ± 2.4	4.5 ± 2.5	<0.001
Unaffected side	4.4 ± 2.7	4.3 ± 2.6	4.6 ± 2.7	4.2 ± 2.2	0.895
*p*-value ^‡^	<0.001	0.317	0.692	0.706	

Abbreviations: PO, postoperative; F/U, follow-up. * Data are represented as mean ± standard deviation. ^†^ Comparison between preoperative and final follow-up. ^‡^ Comparison between affected (operated) and unaffected sides.

**Table 4 jcm-14-01683-t004:** The changes in static postural control ability (Wilcoxon signed-rank test).

Modified Romberg Test	Preoperative	PO 6 Months	PO 1 Year	Final F/U	*p*-Value ^†^
Balance retention time (sec) *					
Affected (operated) side	3.7 ± 2	4.6 ± 2.4	5.7 ± 2.8	6.4 ± 2.9	<0.001
Unaffected side	7.6 ± 3.5	7.8 ± 3.8	7.3 ± 3.4	7.5 ± 3.3	0.975
*p*-value ^‡^	<0.001	<0.001	0.002	0.105	

Abbreviation: PO, postoperative; F/U, follow-up. * Data are represented as mean ± standard deviation. ^†^ Comparison between preoperative and final follow-up. ^‡^ Comparison between affected (operated) and unaffected sides.

**Table 5 jcm-14-01683-t005:** The changes in dynamic postural control ability (Wilcoxon signed-rank test).

Biodex Posturography	Preoperative	PO 6 Months	PO 1 Year	Final F/U	*p*-Value ^†^
Overall stability index *					
Affected (operated) side	3.7 ± 2.1	3.1 ± 1.6	2.2 ± 1.3	1.9 ± 1.1	<0.001
Unaffected side	1.5 ± 0.9	1.7 ± 1.1	1.4 ± 0.9	1.4 ± 0.8	0.935
Deficit ratio	59.5%	45.2%	36.4%	26.3%	
*p*-value ^‡^	<0.001	<0.001	0.001	0.014	

Abbreviation: PO, postoperative; F/U, follow-up. * Data are represented as mean ± standard deviation. ^†^ Comparison between preoperative and final follow-up. ^‡^ Comparison between affected (operated) and unaffected sides.

## Data Availability

The data presented in this study are available in the article.

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
