# Peer review of "The Modified Broström Procedure with Suture-Tape Augmentation for Chronic Lateral Ankle Instability"

_jcm, 2025, doi:10.3390/jcm14051683_

Round 1

Reviewer 1 Report

Comments and Suggestions for Authors

Thank you to the authors for their work in preparing the manuscript. Personally I have an inherent bias here as I believe that suture tape augmentation is an effective technique in most patients undergoing lateral ankle ligament reconstruction. One of the criticisms of this technique is stiffness and potential foreign body reaction, and in those patients where the tape is not placed in the correct anatomic footprint – there is potential for poor outcome. It is good to see a paper with reasonable follow up of this technique.

The paper could be made stronger by having a comparison group (i.e, patients who have a MBP withoutsuturetape augmentation). But I understand of course this may not be possible.

See specific comments below:

Abstract:

-              Nil to add.

Introduction

-              I think it would be worth the authors defining, anatomically what the original and modified brostom operations are, somewhere in the introduction.

-              Lines 43-44: what are the poor prognostic factors leading to poor outcomes in MBP – I think worth stating these with references

-              Is “suture tape” a trade name (Arthrex) – I would double check this, and if so I would change it throughout the manuscript

-              Lines 56-57 – who are these populations?

Materials and methods:

-              It is not clear to the reader, if the surgeon uses the suture tape augmentation in all patients, or just in the ones listed as “indicated” in lines 74-80. Please clarify this.

-              Did any patients have ATFL and CFL augmentation (i.e, did the surgeons ever use anchors in the fibular, talus and calcaneus). This is a recognised technique, but augmentation of the CFL with suture tape is not mentioned in the operative technique section. Please clarify.

-              The figures are very nice – well done

-              Lines 144-145: was there any attempt to measure inter-observer reliability with radiographic measurements?

Results:

-              Nil to add

Discussion:

-              Lines 257-258: I would disagree and I think the results are stronger, I would consider rewording “may be an effective” to “is an effective option”

-              Line 266-268:  is this a formal comparison group to MBP without augmentation? If so then the scores for these patients should presented as a comparison in the tables in the results section. If not then this sentence should be removed – because as far as I could tell – there was no formal comparison group in this paper.

-              I would mention somewhere in the discussion about the applications for patients with poor quality native tissues such as those with collagen disorders, the synthetic ligament overcomes this biological issue. I don’t know of any but are there any papers in the literature looking at ligamentous laxity/Ehler Danlos/Marfan patients ? 

-              Lines 354-357: this is an important point, the ATFL internal brace needs to be placed at the anatomic insertion point of the ligament on the talus. I’m sure there are biomechanical or cadaveric studies which can be cited on this.

-              I would list the lack of formal comparison group as a limitation in this series also.

Conclusions:

-              Line 362: remove “the”

References:

-              No issue

Author Response

Thank you to the authors for their work in preparing the manuscript. Personally I have an inherent bias here as I believe that suture tape augmentation is an effective technique in most patients undergoing lateral ankle ligament reconstruction. One of the criticisms of this technique is stiffness and potential foreign body reaction, and in those patients where the tape is not placed in the correct anatomic footprint - there is potential for poor outcome. It is good to see a paper with reasonable follow up of this technique.

The paper could be made stronger by having a comparison group (i.e, patients who have a MBP without suturetape augmentation). But I understand of course this may not be possible.

→ Thank you for valuable comment. We are going to keep in mind this point for further study design. Because many previous papers have reported the satisfactory results after MBP with suture-tape augmentation for general population, this study focused on the outcomes and complications after surgery in the patients with relatively poor prognostic factors. Like your comment, a definitive conclusion should be answered through a randomized controlled study with more high-level of evidence.

1) Abstract:

Nil to add.

2) Introduction:

I think it would be worth the authors defining, anatomically what the original and modified brostom operations are, somewhere in the introduction.

Author: As your advice, we described additional contents about original and a few modified Broström procedures.

→ Original Broström procedure for lateral ankle instability was a direct anatomical repair that involved suturing the torn or attenuated ligament ends back together. The Broström procedure has since evolved with a variety of modifications. One of the representative modified Broström repair techniques was the Broström-Gould procedure which included augmentation with the inferior extensor retinaculum over the repaired ligaments. Another technique was the Broström-Karlsson procedure which made shortening of the ligaments themselves through reattachment using transosseous tunnels.

Lines 43-44: what are the poor prognostic factors leading to poor outcomes in MBP – I think worth stating these with references

Author: The relatively poor prognostic factors for the conventional MBP include generalized ligamentous laxity, prior unsuccessful ligament repair surgery, poor quality of remnant ligamentous tissue, large subfibular ossicle, high obesity, high-demand athletes, et al. As your advice, we added a few references.

â‘  DiGiovanni CW, Brodsky A. Current concepts: lateral ankle instability. Foot Ankle Int. 2006;27:854-866.

â‘¡ Kobayashi T, Gamada K. Lateral ankle sprain and chronic ankle instability: a critical review. Foot Ankle Spec. 2014;7:298-326.

â‘¢ Lan R, Piatt ET, Bolia IK, Haratian A, Hasan L, Peterson AB, et al. Suture tape augmentation in lateral ankle ligament surgery: current concepts review. Foot Ankle Orthop. 2021;6: 24730114211045978.

Is “suture tape” a trade name (Arthrex) – I would double check this, and if so I would change it throughout the manuscript

Author: InternalBrace® is the product name (Arthrex, Naples, FL, USA) and suture-tape is one of the components, such as suture anchor. This technique is commonly referred to as suture-tape augmentation.

Lines 56-57 – who are these populations?

Author: 'Various populations' means both patients with and without relatively poor prognostic factors (generalized ligamentous laxity, prior unsuccessful repair surgery, poor quality of remnant ligamentous tissue, high obesity, high-demand athletes, et al) for the conventional MBP.

3) Materials and methods:

It is not clear to the reader, if the surgeon uses the suture tape augmentation in all patients, or just in the ones listed as “indicated” in lines 74-80. Please clarify this.

Author: As your advice, we edited this paragraph to clarify the indications of suture-tape augmentation used in the current study.

→ The MBP augmented with suture-tape was limitedly indicated for the selected patients as followings; 21 patients with generalized ligamentous laxity (Beighton score ≥ 5 points), 11 patients with failed previous ligament repair surgery, 27 patients with poor quality of the remnant lateral ligamentous tissue as shown by preoperative magnetic resonance imaging (MRI), 39 patients with insufficient repairable ligamentous tissue at intraoperative inspection, 12 patients with large subfibular ossicle (diameter ≥ 5 mm), and 9 patients with hyper-obesity (body mass index ≥ 30 kg/m2).

Did any patients have ATFL and CFL augmentation (i.e, did the surgeons ever use anchors in the fibular, talus and calcaneus). This is a recognised technique, but augmentation of the CFL with suture tape is not mentioned in the operative technique section. Please clarify.

Author: Suture-tape augmentation for both ATFL and CFL may be more effective and technically not difficult. Unfortunately, we have no case with suture-tape augmentation for both ATFL and CFL. As your advice, we edited this paragraph to clarify a surgical technique used in the current study.

→ In summary, surgical technique used in this study was an anatomical repair of both ATFL and CFL, and additional suture-tape augmentation for the repaired ATFL.

The figures are very nice - well done

Lines 144-145: was there any attempt to measure inter-observer reliability with radiographic measurements?

Author: The reliability of stress radiological measurements performed in this study was evaluated by calculating intraclass correlation coefficients, which indicated interobserver agreement and intraobserver reproducibility. Intraobserver reproducibility of three researchers was average 0.94 (range, 0.92–0.97) and interobserver agreement between three researchers was average 0.92 (range, 0.88–0.95). Therefore, the reliability of stress radiological measurements was found to be excellent.

4) Results:

Nil to add

5) Discussion:

Lines 257-258: I would disagree and I think the results are stronger, I would consider rewording “may be an effective” to “is an effective option”

Author: As your advice, we made the changes.

→ The most important finding of this study is that MBP with suture-tape augmentation is an effective surgical option for chronic lateral ankle instability with poor prognostic factors related to the conventional anatomical ligament repair.

Line 266-268:  is this a formal comparison group to MBP without augmentation? If so then the scores for these patients should presented as a comparison in the tables in the results section. If not then this sentence should be removed – because as far as I could tell – there was no formal comparison group in this paper.

Author: We are sorry to make a confusion. Based on the clinical results in previous studies reported from our institute, we stated this sentence. As your advice, we removed this sentence.

I would mention somewhere in the discussion about the applications for patients with poor quality native tissues such as those with collagen disorders, the synthetic ligament overcomes this biological issue. I don’t know of any but are there any papers in the literature looking at ligamentous laxity/Ehler Danlos/Marfan patients ? 

Author: In 2017, we reported a study investigating the clinical outcomes after suture-tape augmentation for chronic ankle instability with generalized ligamentous laxity (joint hypermobility). To

our knowledge, this article was the first clinical report regarding the issue which you point out, and suture-tape augmentation in addition to the modified Broström procedure appeared to be an effective operative alternative for chronic ankle instability with generalized ligamentous laxity. As your advice, we described additional contents about the applications of suture-tape augmentation for patients with poor quality connective tissue in Discussion section.

â‘  Cho BK, Park KJ, Park JK, SooHoo NF. Outcomes of the modified Broström procedure augmented with suture-tape for ankle instability in patients with generalized ligamentous laxity. Foot Ankle Int. 2017;38(4):405-411.

→ Due to the inherent connective tissue abnormality, generalized ligamentous laxity has been generally known to be a poor prognostic factor for anatomic ligament repair. Park et al. have reported that generalized ligamentous laxity is an independent predictor of poor outcomes with high rates of clinical failure. Xu et al. suggested that additional augmentation procedure or ligament reconstruction using tendon-graft should be considered as a primary surgery for patients with high Beighton scores (≥7 points). Under a hypothesis that the synthetic ligament such as suture-tape would be able to overcome these biological issue, Cho et al. investigated the clinical outcomes after suture-tape augmentation for chronic ankle instability with generalized ligamentous laxity (joint hypermobility), and

reported that suture-tape augmentation in addition to the MBP appeared to be an effective operative alternative for patients with inherent connective tissue deficiency.

â‘  Baumhauer JF, O’Brien T. Surgical consideration in the treatment of ankle instability. J Athl Train. 2002;37:458-462.

â‘¡ Park KH, Lee JW, Suh JW, Shin MH, Choi WJ. Generalized ligamentous laxity is an independent predictor of poor outcomes after the modified Broström procedure for chronic lateral ankle instability. Am J Sports Med. 2016;44(11):2975-2983.

â‘¢ Xu HX, Lee KB. Modified Broström procedure for chronic lateral ankle instability in patients with generalized joint laxity. Am J Sports Med. 2016;44(12):3152-3157.

Lines 354-357: this is an important point, the ATFL internal brace needs to be placed at the anatomic insertion point of the ligament on the talus. I’m sure there are biomechanical or cadaveric studies which can be cited on this.

Author: We are sorry that we could not find appropriate literatures for this issue. We added a sentence describing necessity of further biomechanical or cadaveric studies regarding the best entry-point of suture-tape and anchors.

→ Although internal brace for the ATFL needs to be placed at the anatomic insertion sites on the talus and fibula, there is still a paucity of literatures on clinical relevance according to the technical divergency. Further biomechanical or cadaveric studies regarding the best entry-point of suture-tape and anchors need to be performed.

I would list the lack of formal comparison group as a limitation in this series also.

Author: As your advice, we described the lack of formal comparison group as another limitation in this study.

→ Finally, the lack of formal comparison group is another limitation in this study. A definitive conclusion regarding usefulness of MBP with suture-tape augmentation in the patients with relatively poor prognostic factors should be answered through a randomized controlled study with more high-level of evidence.

6) Conclusions:

 Line 362: remove “the”

Author: As your advice, we made the changes.

7) References:

No issue

Reviewer 2 Report

Comments and Suggestions for Authors

General Impression 

This study investigated the clinical outcomes of patients with chronic lateral ankle instability who underwent the modified Broström procedure with suture-tape augmentation. The assessment encompassed FAOS, FAAM, static and dynamic postural control ability tests, and stress radiography. The methodology of this study is rigorous, and the presentation of the Materials and Methods and Results sections is well-executed. Overall, this is a commendable paper. 

Abstract 

No specific comments. 

Introduction 

Line 66: “We prospectively investigated ...” It is unclear whether this study was prospective or retrospective. Please refer to the comment regarding this issue below. 

Materials and Methods 

Lines 84-85: Please add an explanation in the text as to why patients with previous ankle ligament surgery were excluded, despite being among the patients with poor prognostic factors. 

Lines 80-81: “Of these patients, ... enrolled in the current study” 

Lines 91-92: “Overall research protocol ... Institutional Review Board” 

The study's prospective or retrospective nature is unclear. Please add information regarding when this study was approved by the IRB. 

Lines 118-122 (Figure 2): This figure may be inappropriate since patients with previous ankle ligament surgery were excluded from this study. Please address this issue. 

Results 

No specific comments. 

Discussion 

No specific comments. 

Conclusions 

No specific comments. 

Author Response

This study investigated the clinical outcomes of patients with chronic lateral ankle instability who underwent the modified Broström procedure with suture-tape augmentation. The assessment encompassed FAOS, FAAM, static and dynamic postural control ability tests, and stress radiography. The methodology of this study is rigorous, and the presentation of the Materials and Methods and Results sections is well-executed. Overall, this is a commendable paper.

1) Abstract

No specific comments.

2) Introduction 

Line 66: “We prospectively investigated ...” It is unclear whether this study was prospective or retrospective. Please refer to the comment regarding this issue below. 

Author: We are sorry to make a confusion. We prospectively collected patient's data and retrospectively analyzed. All evaluation factors were determined during the study design phase, and patients who underwent surgery were enrolled consecutively. All patients were given a written consent regarding the promised visit frequency and examination content after surgery. Finally, patients with follow-up longer than 3 years postoperatively were analyzed in this study.

→ As your advice, we deleted ‘prospectively’.

3) Materials and Methods 

Lines 84-85: Please add an explanation in the text as to why patients with previous ankle ligament surgery were excluded, despite being among the patients with poor prognostic factors. 

Author: As your advice, we described additional contents about why patients with previous ankle ligament surgery were excluded from this study. In most studies evaluating the clinical outcomes after surgical treatment for CAI patients, it was also taken into account that patients with previous surgery history involving the ankle ligaments were excluded from the study subjects.

→ With consideration for heterogeneity in details of the surgical technique (range of skin incision, number and ingredient of remained suture anchors, insertion-point of suture anchors, use of the CF ligament or inferior extensor retinaculum, et al) and remnant ligamentous tissue condition (presence of iatrogenic scar tissue adhesion and remained suture materials), we determined a history of previous ankle ligament surgery as one of the exclusion criteria.

Lines 80-81: “Of these patients, ... enrolled in the current study” 

Author: As your advice, we made the changes.

Lines 91-92: “Overall research protocol ... Institutional Review Board”

The study's prospective or retrospective nature is unclear. Please add information regarding when this study was approved by the IRB.

Author: As your advice, we added information regarding when this study was approved by the IRB.

→ For this study, we prospectively collected patient's data and retrospectively analyzed. All evaluation factors were determined during the study design phase, and patients with eligibility for inclusion criteria were enrolled consecutively. Overall research protocol was approved by the ethics committee of Institutional Review Board at September 2017 (prior to enrollment of the first participant who underwent operation). Informed consent regarding the promised visit frequency and examination content after surgery was given to all participants.

Lines 118-122 (Figure 2): This figure may be inappropriate since patients with previous ankle ligament surgery were excluded from this study. Please address this issue.

Author: We are sorry to make a confusion. We edited Figure 2.

4) Results

No specific comments.

5) Discussion

No specific comments.

6) Conclusions

No specific comments.
